# Sustainable Knowledge Transfer from Business Simulations to Working Environments: Correlational vs. Configurational Approach

Daniel Lovin, Monica Raducan *, Alexandru Capatina and Nicoleta Cristache

Department of Business Administration, Faculty of Economics and Business Administration, Dunarea de Jos University of Galati, 800001 Galati, Romania; daniel.lovin@ugal.ro (D.L.); alexandru.capatana@ugal.ro (A.C.); nicoleta.cristache@ugal.ro (N.C.)
* Correspondence: monica.raducan@ugal.ro

**Abstract:** Employing both a correlational and a configurational framework, this study proposes that engagement in business simulations, working environment culture, and acquired knowledge on business simulations are forerunners of sustainable knowledge transfer from business schools to organizations through business simulations training. Using a sample of 120 graduates from a Romanian business school, the results from configurational framework (based on regression analyses) reveal that knowledge transfer is explained by engagement in business simulations and working environment culture. However, findings highlight no correlation between acquired knowledge through business simulations and knowledge transfer. We have also employed fsQCA (fuzzy-set qualitative comparative analysis), which reveals that engagement in business simulations, working environment culture, and acquired knowledge on business simulations are adequate conditions for knowledge transfer. This study sheds light on a new research avenue of knowledge transfer from business schools to industry, less investigated by prior research.

**Keywords:** business simulation; knowledge acquisition; knowledge transfer; experiential learning

## 1. Introduction

Knowledge transfer from business schools to industry is a stream of research that has attracted researchers in the past several years. Business schools are perceived as knowledge intensive organizations due to their powerful dynamics of knowledge creation and transfer to business organizations. Knowledge transfer from the academic environment to business organizations is affected the asymmetric distribution of knowledge, while intergenerational knowledge transfer is turning into an important process at the academic level [1]. One key pillar of the sustainable knowledge transfer between business schools and private organizations is represented by a greater reliance on intangible resources and on intellectual competences of using them [2].

Knowledge transfer from business schools to work environments through serious games have gained much attention of academic scholars and business community of practice. However, knowledge transfer in the context of simulation game learning still has to be empirically examined.

The focus of this study was on the transfer of knowledge from business schools to organizations through business simulations, perceived as experiential learning methods that enable sustainable knowledge transfer. Knowledge that students acquire from business schools needs to be transferred to organizations where they will be hired. Although the literature on knowledge transfer from business schools to industry has outlined the precursors of knowledge transfer in "Triple Helix" of university-industry-government relationships [3], the following research question arises: Which are the determinants of the knowledge transfer from business schools to organizations through business simulations?

This study attempts to answer the above research question by examining three variables affecting knowledge transfer from business schools to organizations through business simulations: engagement in business simulations, working environment culture, and acquired knowledge on business simulations.

The remainder of the paper highlights the theoretical background and hypotheses development, correlational and configurational frameworks, data analysis and findings, discussion, conclusions implications, limitations, and further research.

## 2. Theoretical Background and Hypotheses

In this research, students' engagement in a business simulation, the knowledge that they acquire during this experiential learning method and the working environment culture from the companies they are working for represent three main factors affecting the transfer of knowledge from business simulations to business organizations.

### 2.1. Knowledge Transfer from Business Schools to Business Organizations

Nonaka [4] highlights the differences between explicit and implicit knowledge. Explicit knowledge is codified and can easily be transferred from business schools to business organizations, whereas implicit knowledge has distinctive traits which are difficult to be transferred. Agrawal [5] explores characteristics of the various channels through which knowledge is transferred from the academic institutions to business organizations, considering the value of sustainable relationships between private companies and universities. The high interest in university–industry relationships for knowledge transfer is mainly based on the universal belief that collaborative approach of innovation within these joint collaborations provides valuable opportunities for both parties [6].

A study conducted by Perkmann and Salter [7] provides a comprehensive analysis of relationships between university–industry, which foster the knowledge transfer, based on a wide range of motives. Knowledge transfer takes place between academic institutions and business organizations on an ongoing basis and the success of knowledge sharing depends on engagement of these entities involved in the transfer process [8].

During their studies, students enrolled in business schools acquire knowledge and skills from practical courses and business simulations. The degree of knowledge acquired from business simulations and other interactive teaching and learning methods represents their ability for knowledge transfer [9]. Students, while working in different companies, effectively apply the knowledge and skills acquired within business simulations to their tasks in their business organizations. The effectiveness of knowledge transfer may be attributed to the motivation to apply the knowledge acquired within business simulations, in a financial free-risk environment, to enhance their job performance [10]. The organizational culture of the organizations where business school graduates are working generates an enabler or inhibitor to transfer their tacit knowledge into their job. The higher the degree of innovation and creativity embedded in the organizational culture, the higher the ability of a business school graduate to transfer tacit knowledge and willingness to use it in his tasks [11]. Business schools' professors should reconsider their teaching methods, and students should embrace active learning models by focusing more on acquiring developing generic thinking skills, rather than on the traditional way of knowledge transfer [12].

The classical approach of teaching and learning in business schools, mostly based on knowledge transfer, becomes obsolete, since the business knowledge lifecycle is shortening and new type of jobs, with new knowledge requests, appear in economy due to digital transformation [13]. The knowledge as energy metaphor, as stated by Bratianu and Bejinaru [14], is highly valuable in understanding knowledge transfer between business schools and companies where graduates are enrolled, as each form of tacit knowledge acquired during faculty can be transformed into another form of knowledge in practical activities and knowledge dynamics means knowledge transformation in specific work environments.

### 2.2. Engagement in Business Simulations—Effects on Knowledge Transfer

Previous studies have proven that business schools' students are motivated in a little extent by traditional courses, finding engagement only in simulation-based settings [15,16]. Therefore, in the business school management teams' quest to solve the problems associated with students' lack of motivation and engagement, researchers have suggested that gamification-based simulations make academic activities more interesting and appealing to students [17] and stimulates knowledge transfer among teams [18].

An interactive teaching method of business concepts that motivates and engages students in an active manner and provides hands-on and minds-on experiences is represented by business simulation games [19]. Business simulation games are based on virtual scenarios of real business situations that empower students to manage a company in a financial risk-free environment and share knowledge among different departments. The core values of business simulation games reveal the capability to develop business students' innovative skills, motivational abilities, and meaningful tasks [20]. The business game-based learning approach enhances positive experiences for students and ultimately improve their learning levels [21].

Business simulation games are considered effective tools enabling students to acquire and develop both hard and soft skills that are highly valued in the business world [22], facilitating the knowledge transfer between business schools and business organizations. The development of generic competences, such as business knowledge acquisition and analysis, decision-making within uncertainty, communication skills, and leadership skills, is highly appreciated by employers who provides jobs to business schools students and graduates [23]. Usability of a simulation game has an impact on learning outcomes in a wide range of sectors [24], including business education. Gaming promotes challenges for students and stimulates their interest to apply in the work environment the learnings [25].

Researchers have proposed a three-dimensional engagement in business simulations model including behavior, emotion, and cognition [26,27]. Cognitive engagement refers to understanding the scenarios and relationships between the constructs embedded in the business simulation, and emotional engagement reflects the feelings associated to learning experience of students immersed into the business simulation, while behavioral engagement outlines the efforts to meet the business simulation goals. Knowledge transfer capability has been associated these three dimensions of engagement [28]. The findings of a recent study conducted by Buil et al. [29] highlight that students' perceptions of competence and autonomy, while being involved in business simulation games, have a positive impact on their cognitive, emotional and behavioral engagement.

Thus, the first hypothesis arises:

**Hypothesis 1 (H1).** *Engagement in business simulations has a positive impact on knowledge transfer.*

### 2.3. Acquired Knowledge from Business Simulations

During their studies in business schools, students acquire knowledge and skills from a wide range of activities, the most relevant being those focused on active learning, such as role play, case studies, hackathons, workshops with entrepreneurs, and business simulations. Several factors may contribute to the level of knowledge and skills acquired, such as teaching professors' capabilities to deliver courses with practical relevance, students' engagement in business simulations, and their absorptive capacity of business concepts. Students and graduates' absorptive capacity outlines their ability to untap job opportunities based on their acquired knowledge and to recognize the value of tacit knowledge and apply it in the work environments. Students are willing to acquire novel knowledge from business simulations that is applicable to their projects or jobs. For that reason, the more knowledge they have acquired from the business simulation, the more opportunities to transfer the knowledge into their projects and jobs arise [30].

Engagement reflects students' predisposition to anticipate situations that require strategic thinking in business simulations, to solve problems associated to different scenarios and make effective decisions. Moreover, the business simulation process reinforces knowledge acquisition through instructors' feedback after the results, allowing students to reflect on and improve their decisions [31].

Lovelace et al. [32] investigated the impact of the use of business simulations on the acquired knowledge related to critical thinking skills, to purposefully assess situations and improve their strategy. They found a positive relationship between use of business simulations and the acquired knowledge on critical thinking skills. Measures of business simulation games effectiveness in acquired knowledge may include overall team performances or potentially an analysis of deeper types of learning or job performance after simulation completion [33].

Accordingly, we hypothesize that:

**Hypothesis 2 (H2).** *Engagement in business simulation has a positive impact on the acquired knowledge.*

The confidence perceived by students on the knowledge and skills they acquire during business simulations and other active learning methods is important for the effectiveness of knowledge transfer by business schools to employers (business organizations) [34]. Employers are calling for the development of skills acquired in business simulations in real work scenarios. They perceive soft skills as prerequisites to perform hard skills and as professional aptitudes that involve knowledge transfer [35]. A study conducted by Brown et al. [36] investigates students' overall acceptance of business simulations as a strategy for knowledge transfer from business theory to practice and highlights that simulations are highly tailored for preparing them for employment through the development of specific skills.

Thus, we hypothesize that:

**Hypothesis 3 (H3).** *Acquired knowledge from business simulations has a positive impact on knowledge transfer.*

### 2.4. Working Environment Culture

Any corporate culture's ambition is to make its employees to adhere to its values, providing the opportunities to apply the knowledge and skills acquired from business schools to their tasks. Thus, the business organizations seek to create a work environment able to transfer knowledge from several possible sources, including learnings from business simulations and other practical courses.

Employees' expertise requires deep knowledge and understanding in a certain field, which is gained through experience, training, and education in business schools. Learning through knowledge transfer enables the interpretation of relevant knowledge acquired in business schools, offering the opportunity to ascertain what information is necessary in decision-making process [37]. From a holistic perspective, working environment culture can be perceived in terms of an ongoing process of identity building through knowledge acquisition and sharing among members, enabling the sensemaking process to cope with organizational challenges [38].

Simulations bridge the gap between the business training and the real world by providing experience with complex issues, involving innovation and creativity. Business simulations stimulate a working environment culture turned into a supporter of knowledge transfer [39]. The shared values in a company concern dominant organizational attributes, leadership styles, organizational mechanisms, and strategic vision focused on knowledge transfer from business schools to peculiar market situations [40]. A corporate culture based on shared values and innovation is an antecedent of knowledge transfer. Therefore, we assume that:

**Hypothesis 4 (H4).** *Work environment culture has a positive impact on knowledge transfer.*

A work environment which supports an innovative culture will give a myriad of opportunity for employees within the organization. Such an organizational climate will foster a learning environment that encourage its employees to invest in developing their decisional skills through business simulations [41]. Executive education leads to a valuable experience that facilitates the transfer of learnings back into the organization. Managers of business organizations should be aware that they need to provide training opportunities to their employees to acquire knowledge and skills from business schools' executive programs to improve their capabilities [42].

In this context, fifth hypothesis arises:

**Hypothesis 5 (H5).** *Work environment culture has a positive impact on acquired knowledge.*

## 3. Research Methods

### 3.1. Research Frameworks

Romanian business schools must cope with the increase of employers' expectations regarding their graduates' hard and soft skills, by enhancing the quality of their educational programs. Romanian business organizations have recognized the role of business knowledge transfer in the highly environment. They expect from busines schools to prepare the workforce in a way that enable the value of acquired knowledge in experiential learning methods, such as business simulations.

In this context, we propose two research frameworks. The first captures the hypotheses previously developed in a correlational model (Figure 1).

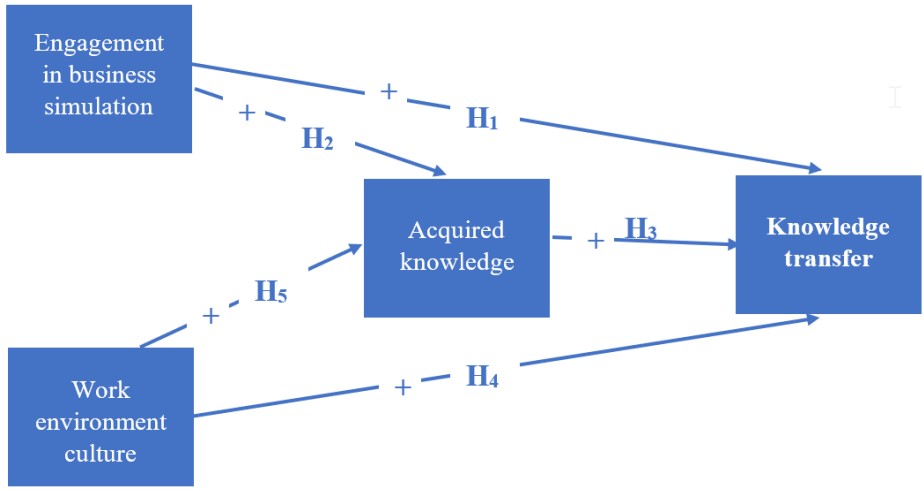

**Figure 1.** Correlational research framework. Source: original contribution.

In the correlational framework, our aim was to assess the statistical relationships between the variables, considering three independent variables: engagement in business simulations, work environment culture, and acquired knowledge, as well as one dependent variable: knowledge transfer.

The correlational research framework is not able to capture the principles of causal complexity, addressed by Qualitative Comparative Analysis (QCA) [43]. QCA is a configurational method that highlights three principles of causal complexity. First, it is primarily used to analyze how multiple, independent causal conditions are combined to affect a given outcome. Second, it helps researchers to assess whether there are different combinations of antecedent conditions associated with the outcome. Third, it explores the possibility that both the presence and the absence of causal conditions could be associated with the outcome [44].

The second framework provides a configurational view on the relationships between the variables embedded in the correlational model, using QCA method, as shown in Figure 2.

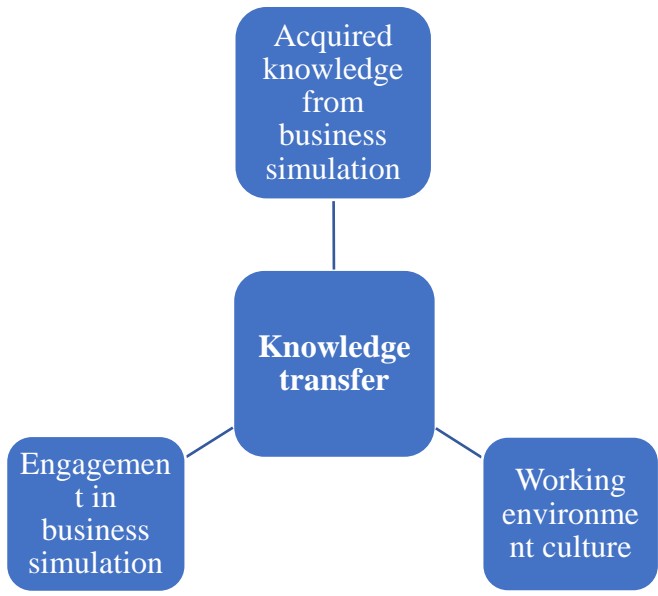

**Figure 2.** Configurational research framework. Source: original contribution.

QCA is considered a useful method to analyze complex causality (effects of combinations between antecedent conditions) and to complement findings from correlational data analysis [45]. The integration of QCA and correlational analysis is a promising way of systematically combining research approaches. The motivations to integrate the results of a QCA in a regression analysis are clearly outlined by Meuer and Rupietta [46]: the opportunity to control for alternative explanations, the opportunity to quantify QCA solutions, to make predictions on the basis of the entire QCA solution, and to test research frameworks with both linear and configurational hypotheses.

### 3.2. Measurements and Research Sample

Four unidimensional constructs were investigated: engagement in business simulations, knowledge transfer, acquired knowledge from business simulation, and work environment culture. Knowledge transfer was measured by four items, revealing the application of knowledge gained from business simulation to respondents' current jobs and the hard and soft skill transferability from business simulations to work environment. Engagement in business simulations was measured by four items, highlighting the role of minds-on and hands-on experiences in business simulations and the level of cognitive, emotional and behavioral engagement. Acquired knowledge from business simulations was measured by four items, outlining the development of problem-solving, creative, teamwork, and leadership skills, while the items related to work environment culture are focused on different corporate values that arises at the organizational level.

All items were measured by a 7-point Likert scale, anchored by 1: strongly disagree, and 7: strongly agree. The items are presented in Table 1.

The scales measuring these constructs were refined via Cronbach's alpha reliability (Cronbach Alpha = 0.871) using the data set collected from 120 graduates from Faculty of Economics and Business Administration, "Dunarea de Jos" University of Galati, representing the research sample. All these graduates participating to this survey have been immersed on active learning methods as business simulations during their studies and were involved in work environments (as full time or part time employees) when they received an invitation to fill in an online survey within Alumni community of the faculty. They responded to the survey in the period October 2020–December 2020.

**Table 1.** Descriptive statistics related to items used in the questionnaire.

| Item | Mean | Std. Deviation | Source |
|---|---|---|---|
| Engagement in business simulations (EBS) | | | |
| EBS1. The hands-on and minds-on experiences keep me engaged during the business simulation. | 5.81 | 1.102 | [47] |
| EBS2. The level of cognitive engagement (understanding the relationships between concepts involved in the business simulation) was high. | 5.83 | 0.816 | [29] |
| EBS3. The level of emotional engagement (feelings related to learning experience during the business simulation) was high. | 5.74 | 0.957 | [29] |
| EBS4. The level of behavioral engagement (effort expenditure to meet the business simulation goals) was high. | 5.73 | 0.968 | [29] |
| Knowledge transfer (KT) | | | |
| KT1. I have applied my knowledge gained from business simulation to my current job. | 5.24 | 0.907 | [48] |
| KT2. I enjoy applying the learnings from business simulation to the tasks assigned in my current job. | 5.43 | 0.866 | [10] |
| KT3. The hard skills developed during the business simulation are transferable to my work environment. | 5.38 | 0.972 | [49] |
| KT4. The soft skills developed during the business simulation are transferable to my work environment. | 5.20 | 1.026 | [49] |
| Acquired knowledge from business simulation (AKBS) | | | |
| AKBS1. The business simulation has developed my problem-solving skills. | 5.83 | 0.827 | [49] |
| AKBS2. The business simulation has developed my creative skills. | 4.30 | 1.142 | [49] |
| AKBS3. The business simulation has developed my ability to work in a team. | 5.61 | 0.998 | [49] |
| AKBS4. The business simulation has developed my leadership skills. | 4.48 | 1.145 | [49] |
| Work environment culture (WEC) | | | |
| WEC1. The company where I am working values each employee contribution to meet the organizational goals. | 4.48 | 1.250 | [9] |
| WEC2. The company where I am working promotes creativity and innovation. | 4.81 | 1.343 | [9] |
| WEC3. The company where I am working values clear procedures and rules. | 4.90 | 1.381 | [9] |
| WEC4. The company where I am working delegates decisions to the lowest hierarchical level. | 4.44 | 1.249 | [9] |

Source: Data processed with SPSS software.

Respondents' demographics is revealed in Table 2. The sample reflects a balanced distribution on respondents' gender. In what concerns their study level, we can also observe a balanced distribution between Bachelor and Master. When the respondents were involved in the study, they declared that they were working on different economic sectors.

**Table 2.** Respondents' demographics.

| Segmentation Criterion | | Number | Percentage |
|---|---|---|---|
| Gender | Male | 48 | 40% |
| | Female | 72 | 60% |

**Table 2.** *Cont.*

| Segmentation Criterion | | Number | Percentage |
|---|---|---|---|
| Study level | Bachelor | 68 | 56.67% |
| | Master | 51 | 42.5% |
| | PhD. | 1 | 0.83% |
| Working sector | Commerce | 31 | 25.83% |
| | Manufacturing | 23 | 19.17% |
| | IT | 36 | 30.00% |
| | Finance-Banking | 19 | 15.83% |
| | Accounting | 11 | 9.17% |

Source: Online survey data.

## 4. Findings

### 4.1. Correlational Approach

Regression analysis have been selected as appropriate statistical analysis to test the hypotheses, as it is able to analyze the relationships between the variables included in the correlational framework and to emphasize the predictors of knowledge transfer from business schools to work environments.

We have passed from linear regression to ANOVA, as we analyzed the same constructs, but presented in different ways. The regression reflects only one mean, and the differences between that one and all other means, and the *p*-values evaluate the comparisons, while ANOVA reports each mean and a *p*-value, highlighting that the variables are significantly different.

**H1.** Engagement in business simulations has a positive impact on knowledge transfer.

The regression analysis output (Table 3) reveals that there is a moderate positive relationship between engagement in business simulations (EBS) and knowledge transfer (KT), as Pearson correlation coefficient R has the value of 0.221. The coefficient of determination ($R^2 = 0.049$) reveals that only 4.9% of the variance in knowledge transfer is explained by engagement in business simulations.

**Table 3.** Regression analysis—first hypothesis.

| | | | | | |
|---|---|---|---|---|---|
| Coefficient of determination | $R^2$ | 0.049 | | | |
| Pearson correlation coefficient | R | 0.221 | | | |
| | Std. Error | 0.617 | | | |
| ANOVA table | | | | | |
| Source | Sum of Squares | df | Mean Square | F | *p*-value (Sig) |
| Regression | 2.307 | 1 | 2.307 | 6.065 | 0.015 |
| Residual | 44.893 | 118 | 0.380 | | |
| Total | 47.200 | 119 | | | |
| Regression output | | | | | |
| Variables | Coefficients | Std. error | t | *p*-value | |
| Constant | 4.088 | 0.495 | 8.252 | 0.000 | |
| Predictor: | | | | | |
| Engagement in business simulations (EBS) | 0.206 | 0.084 | 2.463 | 0.015 | |

Dependent Variable: Knowledge transfer (KT). Source: Data processed with SPSS software.

The ANOVA test highlights that the regression model predicts knowledge transfer significantly because the calculated F ratio of 6.065 is greater than the tabulated F ratio value of 3.92 ($F_{1,119} = 3.92$), and the generated *p*-value (0.015) is lower than threshold

0.05, which is statistically significant at 95% confidence interval. The first hypothesis is supported.

The regression model related to first hypothesis is:

$$KT = \alpha + \beta\ EBS, \tag{1}$$

$$KT = 4.088 + 0.206\ EBS. \tag{2}$$

The results of regression coefficients reveal that engagement in business simulations contributes statistically to the model ($\beta$ = 0.206, t = 2.463, *p* = 0.015) and can be used to predict knowledge transfer (KT). For every additional unit of EBS, KT is expected to increase by an average of 0.206 units.

**H2.** Engagement in business simulation has a positive impact on the acquired knowledge.

The regression analysis output (Table 4) highlights the lack of relationship between engagement in business simulations (EBS) and acquired knowledge (AK), as Pearson correlation coefficient R has the value of 0.010. The coefficient of determination ($R^2$ = 0.0009) reveals that less 1% of the variance in acquired knowledge is explained by engagement in business simulations.

**Table 4.** Regression analysis—second hypothesis.

| Coefficient of determination | $R^2$ | 0.0009 | n | 120 | |
|---|---|---|---|---|---|
| Pearson correlation coefficient | R | 0.010 | | | |
| | Std. Error | 0.560 | | | |
| ANOVA table | | | | | |
| Source | Sum of Squares | df | Mean Square | F | *p*-value (Sig) |
| Regression | 0.003 | 1 | 0.003 | 0.011 | 0.916 |
| Residual | 36.988 | 118 | 0.313 | | |
| Total | 36.992 | 119 | | | |
| Regression output | | | | | |
| Variables | Coefficients | Std. error | t | *p*-value | |
| Constant | 5.042 | 0.486 | 10.379 | 0.000 | |
| Predictor: | | | | | |
| Engagement in business simulations (EBS) | −0.009 | 0.086 | −0.105 | 0.916 | |

Dependent Variable: Acquired knowledge (AK). Source: Data processed with SPSS software.

The ANOVA test outlines that the regression model does not predict acquired knowledge because the calculated F ratio of 0.011 is lower than the tabulated F ratio value of 3.92 (F1,119 = 3.92), and the generated *p*-value (0.916) is higher than threshold 0.05, which is statistically significant at 95% confidence interval. Thus, the second hypothesis is rejected.

The regression model related to second hypothesis is:

$$AK = \alpha + \beta\ EBS, \tag{3}$$

$$AK = 5.042 - 0.009\ EBS. \tag{4}$$

The results of regression coefficients reveal that engagement in business simulations cannot be used to predict acquired knowledge (AK). For every additional unit of EBS, AK is expected to decrease by an average of 0.009 units.

**H3.** Acquired knowledge from business simulations has a positive impact on knowledge transfer.

The regression analysis output emphasized in Table 5 reveals the lack of association between acquired knowledge (AK) and knowledge transfer (KT), as Pearson correlation coefficient R has the value of 0.160, being near zero. The coefficient of determination

($R^2$ = 0.026) supports the statement that less 1% of the variance in knowledge transfer is explained by acquired knowledge.

**Table 5.** Regression analysis—third hypothesis.

| Coefficient of determination | $R^2$ | 0.026 | n | 120 | |
|---|---|---|---|---|---|
| Pearson correlation coefficient | R | 0.160 | | | |
| | Std. Error | 0.671 | | | |
| ANOVA table | | | | | |
| Source | Sum of Squares | df | Mean Square | F | *p*-value (Sig) |
| Regression | 1.398 | 1 | 1.398 | 3.102 | 0.081 |
| Residual | 53.194 | 118 | 0.451 | | |
| Total | 54.592 | 119 | | | |
| Regression output | | | | | |
| Variables | Coefficients | Std. error | t | *p*-value | |
| Constant | 5.779 | 0.554 | 10.423 | 0.000 | |
| Predictor: Acquired knowledge (AK) | −0.194 | 0.110 | −1.761 | 0.081 | |

Dependent Variable: Knowledge transfer (KT).

The ANOVA test confirms that the regression model does not predict knowledge transfer because the calculated F ratio of 3.102 is lower than the tabulated F ratio value of 3.92 ($F_{1,119}$ = 3.92), and the generated *p*-value (0.081) is higher than threshold 0.05. In this context, third hypothesis is also rejected.

The regression model related to third hypothesis is:

$$KT = \alpha + \beta\ AK, \tag{5}$$

$$KT = 5.779 - 0.194\ AK. \tag{6}$$

The results of regression coefficients reveal that acquired knowledge cannot be used to predict knowledge transfer (KT). For every additional unit of AK, KT is expected to decrease by an average of 0.194 units.

**H4.** Work environment culture has a positive impact on knowledge transfer.

The regression analysis output (Table 6) reveals that there is a significant positive relationship between work environment culture (WEC) and knowledge transfer (KT), as Pearson correlation coefficient R has the value of 0.412. The coefficient of determination ($R^2$ = 0.170) proves that 17% of the variance in knowledge transfer is explained by work environment culture.

The ANOVA test reflects that the regression model predicts knowledge transfer significantly because the calculated F ratio of 24.160 is greater than the tabulated F ratio value of 3.92 ($F_{1,119}$ = 3.92), and the generated *p*-value (0.001) is lower than threshold 0.05, which is statistically significant at 95% confidence interval. The fourth hypothesis is statistically supported.

The regression model related to fourth hypothesis is:

$$KT = \alpha + \beta\ WEC, \tag{7}$$

$$KT = 0.720 + 0.815\ WEC. \tag{8}$$

The results of regression coefficients highlight that work environment culture contributes statistically to the model ($\beta$ = 0.815, t = 4.915, *p* = 0.000) and can be used to predict knowledge transfer (KT). For every additional unit of WEC, KT is expected to increase by an average of 0.815 units.

**H5.** Work environment culture has a positive impact on acquired knowledge.

The regression analysis output revealed in Table 7 provides information regarding the lack of relationship between work environment culture (WEC) and acquired knowledge

(AK), as Pearson correlation coefficient R has the value of 0.177. The coefficient of determination ($R^2$ = 0.031) reveals that 3.1% of the variance in acquired knowledge is explained by work environment culture.

**Table 6.** Regression analysis—fourth hypothesis.

| Coefficient of determination | $R^2$ | 0.170 | n | 120 | |
|---|---|---|---|---|---|
| Pearson correlation coefficient | R | 0.412 | | | |
| | Std. Error | 0.620 | | | |
| ANOVA table | | | | | |
| Source | Sum of Squares | df | Mean Square | F | *p*-value (Sig) |
| Regression | 9.278 | 1 | 9.278 | 24.160 | 0.001 |
| Residual | 45.314 | 118 | 0.384 | | |
| Total | 54.592 | 119 | | | |
| Regression output | | | | | |
| Variables | Coefficients | Std. error | t | *p*-value | |
| Constant | 0.720 | 0.834 | 0.863 | 0.390 | |
| Predictor: Work environment culture (WEK) | 0.815 | 0.166 | 4.915 | 0.000 | |

Dependent Variable: Knowledge transfer (KT).

**Table 7.** Regression analysis—fifth hypothesis.

| Coefficient of determination | $R^2$ | 0.031 | n | 120 | |
|---|---|---|---|---|---|
| Pearson correlation coefficient | R | 0.177 | | | |
| | Std. Error | 0.551 | | | |
| ANOVA table | | | | | |
| Source | Sum of Squares | df | Mean Square | F | *p*-value (Sig) |
| Regression | 1.155 | 1 | 1.155 | 3.804 | 0.054 |
| Residual | 35.837 | 118 | 0.304 | | |
| Total | 36.992 | 119 | | | |
| Regression output | | | | | |
| Variables | Coefficients | Std. error | t | *p*-value | |
| Constant | 3.549 | 0.741 | 4.786 | 0.000 | |
| Predictor: Work environment culture (WEK) | 0.288 | 0.147 | 1.950 | 0.054 | |

Dependent Variable: Acquired knowledge (AK).

The ANOVA test confirms that the regression model does not predict acquired knowledge because the calculated F ratio of 3.804 is lower than the tabulated F ratio value of 3.92 ($F_{1,119}$ = 3.92), and the generated *p*-value (0.054) is slightly higher than threshold 0.05, which is statistically significant at 95% confidence interval. Thus, the fifth hypothesis is rejected.

The regression model related to fifth hypothesis is:

$$AK = \alpha + \beta \; WEK, \tag{9}$$

$$AK = 3.549 + 0.288 \; WEK. \tag{10}$$

The results of regression coefficients reveal that work environment culture cannot be used to predict acquired knowledge (AK). For every additional unit of WEK, AK is expected to increase by an average of 0.288 units.

### 4.2. Configurational Approach

Most researchers use QCA approach in a regression-based framework to contrast the correlational findings [50]. Following this trend, this study offers insights on data using fuzzy-set qualitative comparative analysis (fsQCA) software for contrasting the results from these two approaches, correlational and configurational.

Table 8 summarizes the test of the hypotheses.

**Table 8.** Summary of hypotheses testing results.

| Hypothesis | Test |
|---|---|
| H1 | Supported |
| H2 | Rejected |
| H3 | Rejected |
| H4 | Supported |
| H5 | Rejected |

Relationships between the variables are considered asymmetric because respondents' opinions vary. Therefore, alternative combinations of causal conditions can lead to the outcome. In the configurational scenario, we consider the outcome knowledge transfer (KT), while the antecedent conditions are: work environment culture (WEC), engagement in business simulations (EBS), and acquired knowledge on business simulations (AKBS).

We translated the values reflected in 7-point Likert scale used in the questionnaire into fuzzy-set scores ranging from 0.00 to 1.00 (Table 9).

**Table 9.** Calibration of scales.

| Scale Point | Value | Fuzzy-Set Value | Membership |
|---|---|---|---|
| Strongly agree | 7 | 1 | Fully in |
| Agree in a large extent | 6 | 0.84 | More in than out |
| Agree in a less extent | 5 | 0.67 | |
| Neither agree or disagree | 4 | 0.5 | Cross-over (neither in nor out) |
| Disagree in a less extent | 3 | 0.33 | More out than in |
| Disagree in a large extent | 2 | 0.16 | |
| Strongly disagree | 1 | 0 | Fully out |

Source: original contribution.

We established three qualitative anchors for the calibration: an anchor to define full membership, an anchor to define an almost complete lack of membership, and a crossover point.

The complex solution provided by the Quine-McCluskey algorithm (EBS*AKBS*WEC) outlines that a single combination of these three antecedent conditions lead to the outcome: knowledge transfer (KT) (Table 10). The equifinality principle is not supported, as there are not multiple pathways leading to the outcome.

As a concluding remark, we can state that the combination between working environment culture (WEC), engagement in business simulations (EBS), and acquired knowledge on business simulations (AKBS) represents a successful recipe for knowledge transfer.

**Table 10.** Complex solution for the model knowledge transfer (KT) = f(engagement in business (EBS), acquired knowledge from business simulation (AKBS), work environment culture (WEC)).

| Complex Solution | Raw Coverage | Unique Coverage | Consistency |
|---|---|---|---|
| EBS*AKBS*WEC | 0.8340 | 0.8340 | 0.9615 |
| | Solution coverage: 0.8340 | | |
| | Solution consistency: 0.9615 | | |

Source: fuzzy-set qualitative comparative analysis (fsQCA) software output.

## 5. Discussion

Analyzing the knowledge transfer from business schools to organizations through business simulations, this study examined the roles of graduates' engagement in business

simulations, working environment culture, and acquired knowledge on business simulations in knowledge transfer. Using a sample of 120 graduates of a Romanian business school, the findings reported through correlational approach outline that, working environment culture is the most influential predictor of knowledge transfer, while the engagement in business simulations has a moderate influence on knowledge transfer. The acquired knowledge on business simulations has not the expected influence on knowledge transfer. Further, acquired knowledge proved to be influential on the adaptation to work environment culture. With a consistency score of 0.9615, the findings reported through configurational approach revealed that graduates' engagement in business simulations, work environment culture, and acquired knowledge are sufficient conditions for knowledge transfer.

This study fills a gap in knowledge transfer: the transfer of knowledge from business schools to organizations through business simulations and further strengthens the theoretical aspects of the sustainable knowledge transfer. To the best of our knowledge, this research brings interesting insights addressing the role of business simulations in university–industry knowledge transfer, offering avenues for approaching sustainable knowledge transfer, not investigated by prior research.

The results provided by QCA methodology highlight that motivation and engagement in business simulations explain not only the knowledge acquired by students during these forms of experiential learning but also the knowledge transfer. Engagement in business simulations enables the sustainable knowledge transfer implicitly business schools' output performance. This finding is in line with a research conducted by Berbegal-Mirabent et al. [51], which outlines that ed-tech platforms and entrepreneurial education are valuable drivers for knowledge transfer. Highly engaged students and graduates are likely to experiment new ideas in the organizations they are working. Thus, they will adapt easier to work environment culture. This finding is consistent with prior research based on knowledge transfer from business schools to business organizations through in-service training students [9].

The results highlight the important role that engagement in business simulations plays in explaining not only the transfer of knowledge but also the fast adaptation to work environment culture. Students' engagement in business simulations enhances their performance on real work environment through knowledge transfer, as well as their acquired knowledge, because highly engaged students are eager to experiment innovative ideas in their business organizations. Thus, this finding is consistent with prior research based on the relationships between firms and universities relative to performance in terms of knowledge transfer success [5].

## 6. Conclusions, Implications and Further Research

The findings of our study have the purpose to challenge the minds of decision-makers from business schools and organizations to recognize the key-enablers of knowledge acquisition in the first instance, in order to develop policies able to improve the effectiveness of sustainable knowledge transfer.

Theoretically, this study fills a gap in knowledge transfer: the transfer of knowledge from business schools to business organizations through simulation games and further strengthens the theoretical aspects of the role played by simulation games in the active learning process deployed in business schools.

A relevant managerial implication of our study consists of research findings' capability to assist the educational outcomes' stakeholders, especially the companies that will hire students and graduates immersed in business simulations during their studies. First, knowledge can be transferred from business schools to industry through appropriate experiential learning methods as business simulations. Thus, a business organization could encourage its employees to apply their knowledge gained from busines school to their tasks by offering cultural values tailored to employees' expectations. Second,

working environment culture enhances the motivation of busines schools' graduates or students to transfer their knowledge acquired through business simulations.

This study has several limitations. First, both correlational and configurational frameworks were tested only with graduates at one business school from Romania. Graduates from other business schools located in Romania may have different attitudes toward knowledge transfer from academia to industry mediated by business simulations. Another limitation is the convenience sample used in this research, considering only the students who were immersed in organizational work environments and accepted to fill in the questionnaire.

Future research should test the conceptual frameworks in other Romanian business schools in order to enhance its generalizability. Second, this study only examines three key antecedents of the knowledge transfer of from business schools to organizations. Other psychological and social variables focused on experiential learning role in knowledge transfer need to be examined in the future research. Further research will investigate the role of leadership in business simulations, workgroup identification, and shared understanding of business concepts within the knowledge transfer process.

**Author Contributions:** Conceptualization, M.R. and N.C.; methodology, A.C. and D.L.; software, D.L.; validation, N.C. and A.C.; formal analysis, D.L.; investigation, M.R. and D.L.; writing—original draft preparation, A.C.; writing—review and editing, M.R. All authors have read and agreed to the published version of the manuscript.

**Funding:** This research received no external funding.

**Institutional Review Board Statement:** Not applicable.

**Informed Consent Statement:** Not applicable.

**Data Availability Statement:** Not applicable.

**Acknowledgments:** This work is conducted within the project ANTREPRENORDOC, in the framework of Human Resources Development Operational Program 2014–2020, financed from the European Social Fund under the contract number 36355/23.05.2019 HRD OP /380/6/13–SMIS (Code: 123847).

**Conflicts of Interest:** The authors declare no conflict of interest.

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
