# Peer review of "Sustainable Knowledge Transfer from Business Simulations to Working Environments: Correlational vs. Configurational Approach"

_sustainability, doi:10.3390/su13042154_

Round 1
Reviewer 1 Report
Dear Authors,
First of all, let me say that I appreciated to have the chance to review your paper. It has potential but – unfortunately – this potential is hidden due to the numerous weaknesses present in the current version of the manuscript. Below you can find more comments and suggestions that I hope will help you in revising your paper, according to the structure of your paper.
Introduction
- Please, provide references for the following statement: Although the literature on knowledge transfer from business schools to industry has outlined the precursors of knowledge transfer in “Triple Helix” of university-industry-government relationships, the question focused on determinants of the knowledge transfer from business schools to organizations through business simulations remains challenging for Romanian educational stakeholders.
- According to the statement “This study attempts to answer the above research question” you have to provide specific research question(s) that currently is not in the manuscript.
Theoretical background and hypotheses
- Please, note that there are typos in section titles, such as “Theoretical bBackground and Hypotheses” and “Acquired kKnowledge from Business Simulations”.
- Useful references to appropriately develop as well as complete the theoretical background are: Zulfiqar, S., Sarwar, B., Aziz, S., Ejaz Chandia, K. and Khan, M.K. (2019) ‘An analysis of influence of business simulation games on business school students’ attitude and intention toward entrepreneurial activities’, Journal of Educational Computing Research, Vol. 57, No. 1, pp.106–130. Kosa, M., Yilmaz, M., O’Connor, R. and Clarke, P. (2016) ‘Software engineering education and games: a systematic literature review’, Journal of Universal Computer Science, Vol. 22, No. 12, pp.1558–1574. Koivisto, J.M., Haavisto, E., Niemi, H., Katajisto, J. and Multisilta, J. (2016) ‘Elements explaining learning clinical reasoning by playing simulation game’, International Journal of Serious Games, Vol. 3, No. 4, pp.29–43. Mawhirter, D.A. and Garofalo, P.F. (2016) ‘Expect the unexpected: simulation games as a teaching strategy’, Clinical Simulation in Nursing, Vol. 12, No. 4, pp.132–136.
Research methods
- Please, within Figure 1, add + or – as per the hypotheses.
- You have to explain what is: QCA, what is a correlation approach, what is a configurational approach and then, what are the pros and cons of all of them. Moreover, you should report other studies that have implemented the same analysis as to support your choice. A question is: what is the value added of comparing the correlation and configurational approach? It is not clear at all what is the value added, what the second explains that the first does not. Do they explain relationships in a different manner? Which manner? This should go before Figure 2. Indeed, right now, figure 2 is posed there without any accompanying explanation.
- In section 3.2 you should provide also references related to the measurement methods used. You state that “Knowledge transfer was measured by four items, revealing the application of knowledge gained from business simulation to respondents’ current jobs and the hard and soft skill transferability from business simulations to work environment. Engagement in business simulations was measured by four items, highlighting the role of minds-on and hands-on experiences in business simulations and the level of cognitive, emotional and behavioral engagement. Acquired knowledge from business simulations was measured by four items, outlining the development of problem-solving, creative, teamwork and leadership skills, while the items related to work environment culture are focused on different corporate values that arises at the organizational level” but it is not clear how the different items used were derived. Moreover, for Table 1 it should be added a column in which you report the source for each item. Moreover, you must have been implemented a Principal Component Analysis as to verify the loadings of your items on the latent constructs.
- R2 is not correct, it should be R2
- Hypothesis are ‘verified’ or ‘supported’, not ‘validated’.
- You jump from regression to ANOVA, please, explain why.
- Please, add a table in which you sum up which hypothesis have been verified and which do not.
- I have perplexities in the collection of answers from students for that study; moreover, what can they say about a ‘Work environment culture’ stemming from the fact that they are students? This is the main drawback of your study. Moreover, it seems to be that you did not conduct any pilot study before doing the research.
- No information about the sample (explaining why it is suitable for that research design), data collection procedure, and how the questionnaire has been administered are provided. You should also provide a table that sums up the basic socio-demographic features of your sample. Moreover, is the sample size sufficient for your data analysis methods?
- Saying “Regression analysis have been selected as appropriate statistical analysis to test the hypotheses” is too reductive. Please, explain pros of this method. Explain also what are the dependent and independent variables and how they are conceptually related.
Findings
- According to the numerous weaknesses of the methodology section, it is not possible to state if finding are adequate or not.
Discussion, Implications and Conclusions
- The section can be further developed and, consequently, divided. Author may have “Discussion” separated from “Conclusions” in order to provide also more information on the theoretical and practical contributions of the study. Moreover, concerning the “generalizability” of the study, in the limitations and future research venues, could be useful to test the proposed investigation not only in other Romanian business schools, but in other business schools in general. Moreover, you should provide examples of other psychological and social variables that could be interesting to investigate in future studies.
Author Response
Dear reviewer,
Thank you for your valuable suggestions to improve the paper.
We are grateful for your recommendations.
Please find attached a table with our comments on your requirements.
Best regards,
Authors

Reviewer 2 Report
Congratulations to the authors for their interesting article! The article has considerable potential to contribute to the current development of educational sciences and management in the education sector.
The empirical section is really precise and very beneficial for all in the academic environment.
In order to help authors improve the level of their article, the following recommendations can be defined:
- The section of Introduction should be substantially extended. It should contain much more disputative starting points (opinions, trends) in the field.
- The section Discussion, Implications and Conclusion has to be divided into 2 parts: Section of Discussion; Section of Conclusion.
- The separate section of Discussion should be relevantly extended. Comparable studies and works of other research teams in the world should be included and discussed in relation to the results obtained by the authors of the paper.
- Remove small formal imperfection (2. Theoretical b Background and Hypotheses - line 51; 3.2. Measurements and rResearch Sample - line 233).
Good luck!
Author Response
Dear Reviewer,
We are grateful for your valuable suggestions and we have integrated new text (highlighted in blue color) to address all your requirements.
Best regards,
Authors
Round 2
Reviewer 1 Report
Dear Author(s),
I appreciated the work you have done on the revised version of the manuscript. However, there are some more aspects that you need to address:
- After Figure 2, it is not still clear in which way QCA complement findings of correlation analysis.
- You declared “are involved in work environments (as full time or part time employees)”. This is a fundamental passage within your work due to the fact that students are asked to rate a work-related measure. Please, provide more details on that.
- Table 2, No. And % should be shown. Moreover, this table should be commented a bit, now it is without accompanying text.
- As already requested, it has been asked to explain why you pass from regression to ANOVA. Within the method you should explain the consecutio of data analysis and their link with the hypothesis to be verified.
Author Response
Dear Reviewer,
Thank you so much for accepting our work on first revision and providing us new suggestions to improve our paper.
We have addressed all your suggestions (all the changes are highlighted in green color in manuscript):
- The integration of QCA and correlational analysis is an promising way of systematically combining research approaches. The motivations to integrate the results of a QCA in a regression analysis are clearly outlined by Meuer and Rupietta: the opportunity to control for alternative explanations, the opportunity to quantify QCA solutions, to make predictions on the basis of the entire QCA solution, and to test research frameworks with both linear and configurational hypotheses.
- All these graduates participating to this survey have been immersed on active learning methods as business simulations during their studies and were involved in work environments (as full time or part time employees) when they received an invitation to fill in an online survey within Alumni community of the faculty. They responded to the survey in the period October 2020 – December 2020.
- Table 2 has been restructured according to your excellent idea. We have also commented on the sample distribution.
-
We have passed from linear regression to ANOVA as we analyzed the same constructs, but presented in different ways. The regression reflects only one mean, and the differences between that one and all other means, and the p-values evaluate the comparisons, while ANOVA reports each mean and a p-value highlighting that the variables are significantly different.
Thank you again for your valuable contribution to our paper improvement!
Authors